# The Prediction of the Remaining Useful Life of Rotating Machinery Based on an Adaptive Maximum Second-Order Cyclostationarity Blind Deconvolution and a Convolutional LSTM Autoencoder

**DOI:** 10.3390/s24082382

**Published:** 2024-04-09

**Authors:** Yangde Gao, Zahoor Ahmad, Jong-Myon Kim

**Affiliations:** Department of Electrical, Electronic and Computer Engineering, University of Ulsan, Ulsan 44610, Republic of Korea; gaoyangdephd@gmail.com (Y.G.); zahooruou@mail.ulsan.ac.kr (Z.A.)

**Keywords:** remaining useful life, health index analysis, convolutional LSTM autoencoder

## Abstract

The prediction of the remaining useful life (RUL) is important for the conditions of rotating machinery to maintain reliability and decrease losses. This study proposes an efficient approach based on an adaptive maximum second-order cyclostationarity blind deconvolution (ACYCBD) and a convolutional LSTM autoencoder to achieve the feature extraction, health index analysis, and RUL prediction for rotating machinery. First, the ACYCBD is used to filter noise from the vibration signals. Second, based on the peak value properties, a novel health index (HI) is designed to analyze the health conditions for the denoising signal, showing a high sensitivity for the degradation of bearings. Finally, for better prognostics and health management of the rotating machinery, based on convolutional layers and LSTM, an autoencoder can achieve a transform convolutional LSTM network to develop a convolutional LSTM autoencoder (ALSTM) model that can be applied to forecast the health trend for rotating machinery. Compared with the SVM, CNN, LSTM, GRU, and DTGRU methods, our experiments demonstrate that the proposed approach has the greatest performance for the prediction of the remaining useful life of rotating machinery.

## 1. Introduction

With the continued industrialization in society, rotating machinery is used in many fields, such as engines, transportation, and aircraft systems. Rotating machinery environments are severe and involve many reasons for failures. Sensors are used to collect the data from the rotating machinery; signal processing methods are deployed to remove the noise; and deep learning methods are applied as prognostics to assess the rotating machinery for prognostics and health management (PHM) [1,2,3].

There are many prognostic approaches for analyzing the health conditions of rotating machinery. Wavelet decomposition makes use of the time-frequency domain for analyzing machinery degradation, and minimum entropy deconvolution (MED) can use the maximum kurtosis to separate impulsive signatures for early-stage fault signals. However, MED needs to overcome excessive kurtosis [4,5]. To resolve this, the maximum correlated kurtosis deconvolution (MCKD) is based on the correlated kurtosis to improve the filtering performance for vibration signals. From the cyclostationary view, the MCKD does not have the best performance for the statistical processing of signals [5,6]. The maximum second-order cyclostationarity blind deconvolution (CYCBD) is proposed to make use of the maximization of the cyclostationary to address the eigenvalue problem, which can improve the recovery of the denoising of the feature signals. However, the prior cyclic frequency influences the performance. To address these problems, an adaptive maximum second-order cyclostationarity blind deconvolution (ACYCBD) is proposed with a cyclic frequency estimation to select the appropriate filter length to remove the noise and to improve the quality of the feature extraction for the vibration signals [7,8].

After filtering the noise from the bearing degradation, a health index is applied to the instantaneous degradation and an assessment of the rotating machinery for the prognostics and health management (PHM) under different operations. To assess the prognostic and health management with the right measures, based on the margin maximization and support vectors, a support vector machine (SVM) is used to analyze the estimates of the rotating machinery. However, there are some limitations, such as the computation expense, model selection, and unsuitability for large datasets. To address these questions, a least-squares support vector machine (LSSVM) optimizes the health index to find a hyperplane for improving the forecasting [9].

Deep learning methods are popularly used in prognostics for the PHM. A convolutional neural network (CNN) utilizes neural networks to improve the prognostic recognition and can automatically learn the remaining useful life (RUL) estimation of rotating machinery. However, a CNN has the disadvantages of overfitting and exploding gradients that decrease the prediction performance [10,11,12,13,14]. For better prognostics and the prognostics and health management (PHM) of the bearing degradation, an LSTM can use the advantages of its architecture for a long memory of bearing degradation and can address the limits and problems for the prediction of the RUL to achieve superior forecasting. The LSTM involves sequential computations due to the recurrent nature of the network. This limits the extent to which operations can be parallelized, potentially slowing training times [15,16]. To improve the quality of the LSTM, a new model integrates the advantages of CNN and LSTM to address the limits for the RUL prediction of rolling bearings. This method can extract sensing data to monitor health states, preserve these benefits, overcome the overfitting of spatial fluctuations, and achieve efficient and accurate health monitoring [17,18,19,20]. In addition, some autoencoder theories are applied to the CNN and LSTM to improve the prediction performance [21,22,23,24]. A gated recurrent unit (GRU) is derived from an LSTM with a smaller number of gates, which can improve the speed of the RUL prediction, and is a dual-thread GRU (DTGRU), which can utilize parallel GRU layers for stronger prediction [25]. Furthermore, several approach technologies based on transfer learning, meta-learning, and other technologies are presented for predicting the remaining useful life of rolling bearings [26,27,28].

To improve the performance of the LSTM method and the application of the convolutional LSTM method, based on convolutional layers and LSTM, autoencoders can achieve a transform convolutional LSTM to develop a convolutional LSTM autoencoder (ALSTM). This proposed method is designed to transform the convolutional recurrent nature of the network and can achieve a high prediction accuracy for the estimates of the RUL of rotating machinery. The major contributions of this study are as follows.

1. The ACYCBD is used to filter noise signals and to identify the fault features of bearing degradation with a cyclic frequency for vibrating signals. This method can avoid the noise–signal interference of bearing degradation, enhancing the prognostic and health management under different operations. The ACYCBD is successful for the feature extraction regarding bearing degradation.

2. After filtering noise from the bearing degradation signals, based on the peak value properties, a good HI is designed to analyze the key rotating machinery degradation and to measure the health conditions in different situations.

3. To provide better prognostics for health management related to bearing degradation, based on convolutional layers and LSTM, an autoencoder can achieve a transform convolutional LSTM to develop a convolutional LSTM autoencoder (ALSTM) with a superior performance to the LSTM method.

The main content is described as follows. The ACYCBD theory is described in Section 2. In Section 3, a novel HI method is designed to measure the degradation trends of the vibrating bearings. The novel ALSTM is constructed to forecast the bearing degradation in Section 4. The experimental results and their discussions and conclusions are shown in Section 5 and Section 6, respectively.

## 2. The Basic Theory of the ACYCBD Method

In the architecture of the ACYCBD, the envelope harmonic product spectrum (EHPS) can detect the cyclic frequency in the vibration signals for the processing of the CYCBD, as in the blind deconvolution theory [6,7,8]. The input signal x is multiplied with the inverse FIR filter h to compute the source signal $s_{0}$. The process is described as follows:(1)$$s=x\;*\;h=\left(s_{0}\;*\;g\right)\;*\;h\approx s_{0}$$

Here s is the estimated input, g is the impulse response, and the convolution operation is expressed as follows:(2)$$\left\{\begin{array}{c} s[L-1]\\ \vdots \\ s[N-1] \end{array}\right\}=\left[\begin{array}{ccc} x[L-1] & \cdots & x[0]\\ \vdots & \ddots & \vdots \\ x[N-1] & \cdots & x[N-L-2] \end{array}\right]\left\{\begin{array}{c} h[0]\\ \vdots \\ h[L-1] \end{array}\right\}$$
where N represents the length of x, L denotes the length of the inverse FIR filter h, and the optimal inverse filter $h_{0}$ is described as follows:(3)$$\;\;\;\;\;\;\;\;h_{0}=\underset{h}{\mathrm{arg max}}\left[O(h)\right]\;\;\;\;$$

Here, $O\left(h\right)$ represents the objective function of the filtering processing, and a cyclic frequency is accumulated from the period of fault impact $T_{s}$, as follows:(4)$$\alpha =\frac{1}{T_{s}}\;\;$$

The key component assessment is described as second-order cyclostationarity ($ICS_{2}$):(5)$${ICS}_{2}=\frac{\sum_{k>0}{\left|c_{s}^{k}\right|}^{2}}{{\left|c_{s}^{0}\right|}^{2}}\;\;\;$$

With
(6)$$\;\;c_{s}^{k}=\langle {\left|s\right|}^{2},e^{j2\pi k\alpha n}\rangle \;\;\;\;\;\;\;\;$$
(7)$$c_{s}^{0}=\frac{{\left\|s\right\|}^{2}}{N-L+1}\;$$

${ICS}_{2}$ is the objective function of the CYCBD:(8)$$h_{0}=\underset{h}{\mathrm{arg max}}{ICS}_{2}\;\;\;\;\;$$

The eigenvalue algorithm (EVA) is designed to optimize the filter coefficient for optimal filtering, as follows:(9)$$c_{s}^{k}=\frac{EDs}{N-L+1}$$
(10)$$c_{s}^{0}=\frac{s^{H}s}{N-L+1}\;\;$$

With
(11)$$\;\;D=diag\left(s\right)=\left[\begin{array}{ccc} \ddots & & 0\\ & s[l] & \\ 0 & & \ddots \end{array}\right]\;\;\;\;\;\;$$
(12)$$\;\;E=[\begin{array}{ccc} e_{1}\cdots & e_{k}\cdots & e_{K} \end{array}]\;\;$$
(13)$$\;\;e_{k}={\left[\begin{array}{cc} e^{-j2\pi k\alpha (L-1)}\cdots & e^{-j2\pi k\alpha (N-1)} \end{array}\right]}^{T}$$

The ${ICS}_{2}$ function is described as follows:(14)$$\;{ICS}_{2}=\frac{s^{H}D^{H}EE^{H}Ds}{{\left|s^{H}s\right|}^{2}}$$

Finally, the result is calculated:(15)$$\;\;{ICS}_{2}=\frac{h^{H}X^{H}WXh}{h^{H}X^{H}Xh}=\frac{h^{H}R_{XWX}h}{h^{H}R_{XX}h}$$

Here, $R_{XWX}$ is the weighted correlation matrix, $R_{XX}$ is the correlation matrix, and the weight matrix W is as follows:(16)$$W=\frac{D^{H}{EE}^{H}D}{s^{H}s}$$

Furthermore, λ is used to calculate the inverse filter $h_{0}$ equivalent to h as follows:(17)$${\;R}_{XWX}h=R_{XX}h\lambda$$

The architecture of the ACYCBD is shown in Figure 1. The EHPS algorithm is applied to determine the cyclic frequency. The steps of the ACYCBD procedure are as follows.

Step 1: The filter length L, initial filter coefficient h, the convergence criterion $\varepsilon_{0}$, the maximum iteration number $N_{max}$, and the initial parameters are applied to compute the temporary signal S for the next optimal operation.

Step 2: The cyclic frequency is calculated using the EHPS method with the optimal maximum amplitude.

Step 3: According to the brief functions, we compute the weight matrix W, the correlation matrix $R_{XX}$, and the weighted correlation matrix $R_{XWX}$, and we optimize the filtering coefficient to achieve the maximum eigenvalue operation.

Step 4: Step 2 is then repeated with the updated value to calculate the optimal filtering coefficient h that satisfies the requirements of the cyclic operation.

Step 5: The optimal denoising signals are then obtained.

Vibration signals are applied to verify the performance of the ACYCBD method, as shown in Figure 2 and Figure 3. In the raw vibration signals, damage produces mixed signals including noise, which interferes with the performance analysis of the vibration conditions.

The envelope spectrum of the raw signals is described in Figure 2. The fault frequency information is shown in the analysis. The initial fault value is set at 12.5 Hz. The frequency multiplication can be calculated in the interval [12.5, 25, 37.5, 50, 62.5, 75, 87.5, 100, 112.5, 125⋯] Hz. There are distinct peaks at frequencies of 25, 37.5, and 50, which are close to two, three, and four times the frequency of a fault. In Figure 3, the ACYCBD filtering method is utilized to remove the noise and to verify the effectiveness of the envelope spectrum. The result was obtained as a fault frequency of 12.5 Hz and was more obvious in the ACYCBD processing. The frequency multiplication was extracted accurately, as [12.5, 25, 37.5, 50, 62.5, 75, 87.5, 100, 112.5, 125⋯] Hz, which shows that the distinct peak frequencies are one to ten times higher than the 12.5 Hz fault frequency. At the same peak frequency of 12.5 Hz, the amplitude of the raw signal is 0.1946; for the ACYCBD, the amplitude is 0.6278. Other processes were also compared. The amplitude is greater than the raw vibration signals at the same peak frequencies, demonstrating the performance of the ACYCBD filtering. The results show that the ACYCBD has a good filtering performance for health analysis, as in the next step.

## 3. The Proposed Health Index

After the filtering process to use deep learning methods for predicting the bearing degradation, the HI is used for an instantaneous assessment of the degradation and health conditions. The HI plays a vital role in the prediction of the degradation of machinery. A good HI strategy can analyze the key components of the equipment degradation for the vibration of the bearings. In this study, the HI method is derived from the peak value theory. The maximum and minimum values are identified by analyzing the changes of the signal and the input vibration signals $\left(\begin{array}{ccc} x_{1}, & \cdots , & x_{n} \end{array}\right)$, searching for the maximum and minimum values of the n numbers from the input data, calculating the difference values for the maximum $x_{p}$ and minimum $x_{q}$, and then evaluating the change from the input signals. The formula is as follows:(18)$$HI=x_{p}-x_{q}$$

## 4. Convolutional LSTM Autoencoder

In this study, we develop a convolutional LSTM autoencoder to address the transformer problem with convolutional LSTM in forecasting. The proposed method consists of a transformer theory that can transform the convolutional layers and LSTM architecture into encoded features. The model is scalable to the encoded features by a mixed model of convolutional layers and LSTM [21,22,23,24].

In the architecture of the proposed method, the convolutional layer utilizes kernels to achieve a convolution operation, which can filter input data to extract the feature information. The formula is shown as follows:(19)$$\;\;\;\;y_{i}^{l+1}\left(j\right)=K_{i}^{l}\;*\;x^{l}\left(j\right)+b_{i}^{l}$$
where $x^{l}\left(j\right)$ represents the input data, $K_{i}^{l}$ denotes the kernel, $b_{i}^{l}$ is the bias, * represents the dot product, and $y_{i}^{l+1}\left(j\right)$ is the computational result.

Figure 4 depicts the fundamental structure of the LSTM, which consists primarily of a cell, an input gate, a forget gate, and an output gate. The cell controls the flow of information. A forget gate decides which information to remove from a previous state; an input gate decides which useful information to store in the current state; and an output gate decide which information to output in the current state. The architecture makes use of any disadvantages to maintain useful, long-term dependencies for prediction. All formulations are shown below.
(20)$$f_{t}=sigmoid(W_{f}x_{t}+U_{f}h_{t-1}+b_{f})\;\;\;$$
(21)$$\;\;i_{t}=sigmoid(W_{i}x_{t}+U_{i}h_{t-1}+b_{i})\;\;\;$$
(22)$$o_{t}=sigmoid(W_{o}x_{t}+U_{o}h_{t-1}+b_{o})\;$$
(23)$$\tilde{C_{t}}=tanh(W_{c}x_{t}+U_{c}h_{t-1}+b_{c})$$
(24)$$C_{t}=f_{t}⨀C_{t-1}+i_{t}⨀\tilde{C_{t}}\;\;\;\;$$
(25)$$\;\;\;\;h_{t}=o_{t}⨀tanh(C_{t})$$
where the vector $x_{t}$ denotes the input data process; W,U, b are the parameters; $f_{t}$ represents the forget gate activation; $i_{t}$ represents the input gate activation; $o_{t}$ represents the output gate activation; $h_{t}$ represents the hidden state vector; $\tilde{C_{t}}$ represents the cell input activation; and $C_{t}$ represents the cell state vector.

The architecture of the convolutional LSTM autoencoder is shown in Figure 5. The parameters of the convolutional layers and the LSTM of the architecture of the convolutional LSTM autoencoder are described in Table 1.

## 5. The experimental Validation

Data are utilized to verify the proposed performance of the approach (from the Intelligent Maintenance System (IMS) center) [6]. Four Rexnord ZA-2115 double-row bearings are applied in the experiment with an AC motor working at a constant speed of 2000 revolutions per minute (RPM) and a sampling frequency of 20,000 Hz. The load is 6000 lb, and PCB 353B33 High-Sensitivity Quartz ICP accelerometers are used to monitor the vibration data. The experimental platform to verify the comparison with different methods of bearing degradation is shown in Figure 6.

The proposed approach for the prognosis is verified for the degradation of bearings as discussed above. The proposed approach for the prognosis resolves three important issues for the prediction of the RUL: the ACYCBD filtering method is utilized to remove noise and obtain the signal; the new HI is designed to describe the bearing degradation efficiency; and the convolutional LSTM autoencoder mode is designed for the prediction of the RUL estimation. The whole process helps to determine the health of the vibration bearings, as shown in Figure 7.

Three datasets describe the bearing degradation experiment. Each file consists of 20,480 points for 1-second records. A total number of 984 files are saved, and the total samples are processed by filtering methods. In Figure 8a, the ACYCBD method is applied to remove noise from the vibration bearing data and to extract the feature signals.

After the extraction of the signal, the HI is used to show the RUL information, in which each file with 20,480 points is calculated into a sample by the HI and the RMS. In the root mean square (RMS), $RMS=\sqrt{\frac{1}{n}\sum_{i=1}^{n}{\left(x\right)}^{2}}$, where x is each value and n is the number of measurements. As in Figure 8b, the novel HI method can measure the bearing degradation to show the health conditions in different situations. The main conditions consist of normal and failure stages, where a fault onset can be observed at file number 544 by the ACYCBD and HI methods. In the ACYCBD+RMS and RMS, the initial bearing degradation begins at file 730.

The convolutional LSTM autoencoder is designed to transform the nature of the network to achieve a priority performance. To better show the prediction process, the total data of 980 files (removing some useless data) are used for training and testing, chosen randomly at the ratio of 1:1. The model is scalable to large memory, neural networks by a mixture model of the encoded features for bearing degradation as in Figure 8c. The data from 500~980 files is shown for the prediction of degradation. To better forecast the performance compared with the existing methods, the root mean square deviation (RMSE) is used, $RMSE=\sqrt{\frac{1}{n}\sum_{i=1}^{n}{\left(y-\hat{y}\right)}^{2}}$, where y is the desired target, $\hat{y}$ is the predicted value, and n is the number. The RMSE is used to evaluate the prediction performance of a convolutional LSTM autoencoder, an SVM, a CNN, an LSTM, a GRU, and a DTGRU. The RMSE is 0.88 for the SVM, 0.56 for the CNN, 0.54 for the LSTM, 0.44 for the GRU, 0.44 for the DTGRU, and 0.42 for the ALSTM, as shown in Table 2. This shows that the ALSTM has a better forecasting performance. Another two datasets are used to verify the proposed methods in Figure 9 and Figure 10.

As in Figure 9a, all samples are processed by ACYCBD filtering. In Figure 9b, after signal extraction, the HI is used to show the RUL information, where the initial bearing degradation occurs at file 721 file in the ACYCBD and HI methods. But in the ACYCBD+RMS and RMS methods, the bearing degradation starts at file 920. This shows that the proposed method can measure bearing degradation more efficiently. Finally, the convolutional LSTM autoencoder is used to show the prediction process. In Figure 9c, files 600~980 are shown for the prediction of degradation. To better forecast the performance compared with the existing methods, the RMSE is used to evaluate the prediction performance of the convolutional LSTM autoencoder, SVM, CNN, LSTM, GRU, and DTGRU. The RMSE is 0.93 for the SVM, 1.18 for the CNN, 0.43 for the LSTM, 0.56 for the GRU, 0.49 for the DTGRU, and 0.35 for the ALSTM. This shows that the convolutional LSTM autoencoder has a better forecasting performance.

Figure 10a shows the filtering process with the ACYCBD method and the degradation measured by the HI. In Figure 10b, the onset of bearing degradation can be observed at file 791 in the ACYCBD and HI methods. With the ACYCBD+RMS and RMS methods, the bearing degradation begins after file 980. This shows that the proposed method can measure bearing degradation more efficiently. Figure 10c shows files 500~980 for the prediction of degradation. To better forecast the performance compared with the existing methods, the RMSE is used to evaluate the prediction performance of the convolutional LSTM autoencoder, SVM, CNN, LSTM, GRU, and DTGRU. The RMSE is 1.15 for the SVM, 1.80 for the CNN, 0.88 for the LSTM, 0.83 for the GRU, 0.63 for the DTGRU, and 0.52 for the convolutional LSTM autoencoder. This shows that the ALSTM has a better forecasting performance than existing methods.

## 6. Conclusions

In this study, the ACYCBD is used to remove the noise from the vibration signals, and the HI is utilized to measure the bearing degradation of a particular condition. Furthermore, the convolutional LSTM autoencoder is constructed to predict the RUL trends of the degradation of bearings. Per the above description, a new hybrid approach, including a filtering method, HI, and a forecasting model, improves the fault diagnosis and monitors for the health conditions of the rotating machinery. Compared with the ACYCBD, RMS, and LSTM, the proposed method showed the greatest performance in the recognition of faults in rotating machinery.

The ACYCBD is used to filter the noise signals and to identify the fault features of the bearing degradation with the cyclic frequency of the vibrating signals. This method can avoid noise interference to identify bearing degradation, enhancing the prognostic and health management under different operating conditions. The ACYCBD performs well for the feature extraction that indicates bearing degradation.

After processing by the ACYCBD method, based on the peak value properties, an HI is utilized to measure the health condition of the vibrating bearings. The HI can assess the conditions of the bearing degradation, showing greater sensitivity to the RUL than existing methods.

The convolutional LSTM autoencoder can assess the RUL of the bearings and shows better prognostic and health management, showing a greater performance than the SVM, CNN, LSTM, GRU, and DTGRU methods.

In the future, multiple types of sensors will be applied to the RUL prediction of the bearing. Furthermore, 2D vibration images may be used in the feature analysis for the prognosis of the vibration bearings. Furthermore, the denoising step in the filtering method is computationally expensive for mixed vibration signals. As a result, in the future, the suggested approach method should be altered to lower the computing complexity for the forecast. The effectiveness of the proposed approach heavily relies on the quality of the data, and a signal processing method could help alleviate this limitation for data cleaning. The proposed approach is used for the prediction of the RUL of the degradation of real bearings. However, it demonstrates that real-world applicability could be constrained by various factors that impact the degradation of bearings, such as the pressures and speeds for the application. It is essential to evaluate the robustness of the approach across diverse real-world studies in the future.

## Figures and Tables

**Figure 1 sensors-24-02382-f001:**
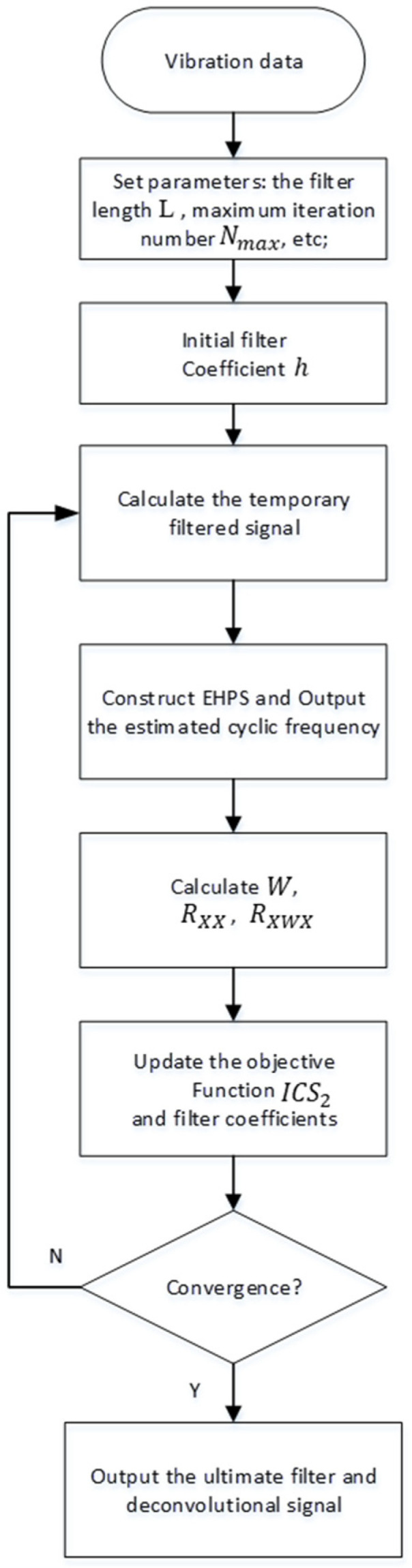
Flowchart of the ACYCBD method.

**Figure 2 sensors-24-02382-f002:**
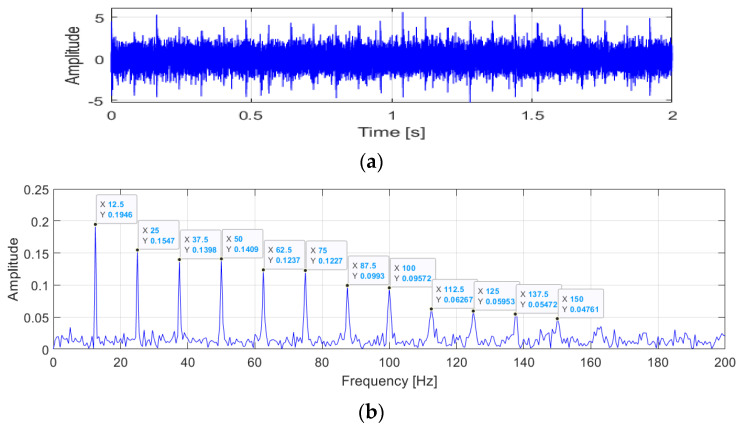
The raw vibration signals: (**a**) signal information in the time domain; and (**b**) the frequency and amplitude of the envelope spectrum.

**Figure 3 sensors-24-02382-f003:**
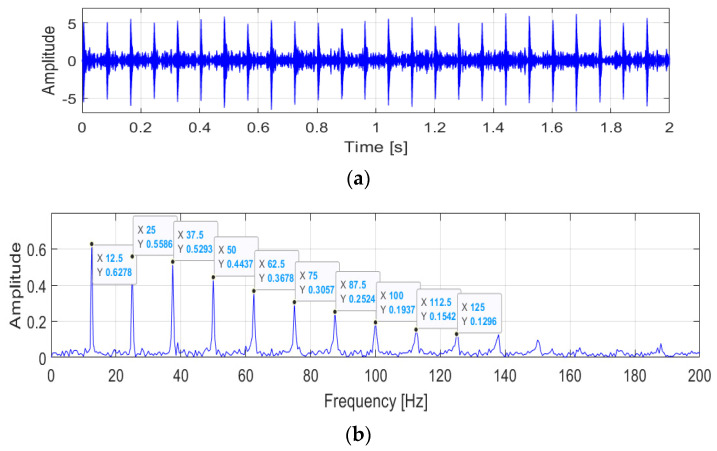
The results obtained by the ACYCBD: (**a**) the denoising signal in the time domain; and (**b**) the frequency and amplitude of the envelope spectrum.

**Figure 4 sensors-24-02382-f004:**
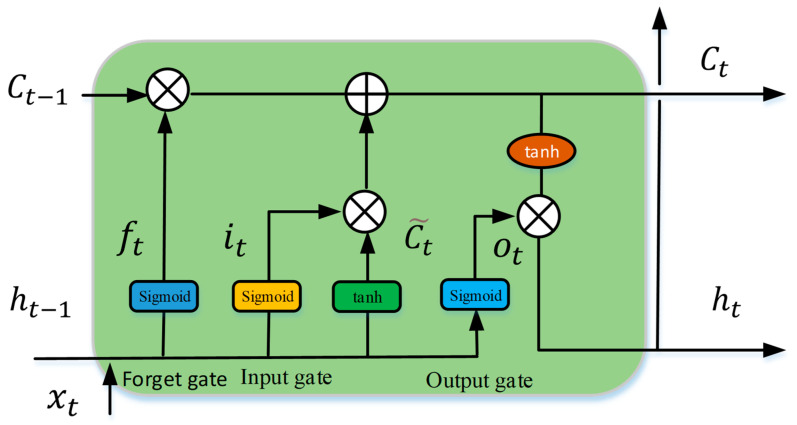
The architecture information of the LSTM.

**Figure 5 sensors-24-02382-f005:**
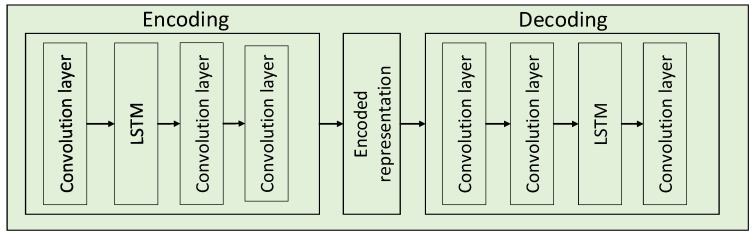
Illustration of the convolutional LSTM autoencoder.

**Figure 6 sensors-24-02382-f006:**
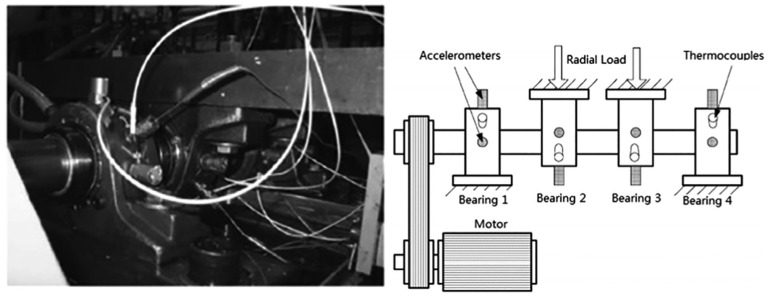
The experimental platform for the degradation of bearings.

**Figure 7 sensors-24-02382-f007:**
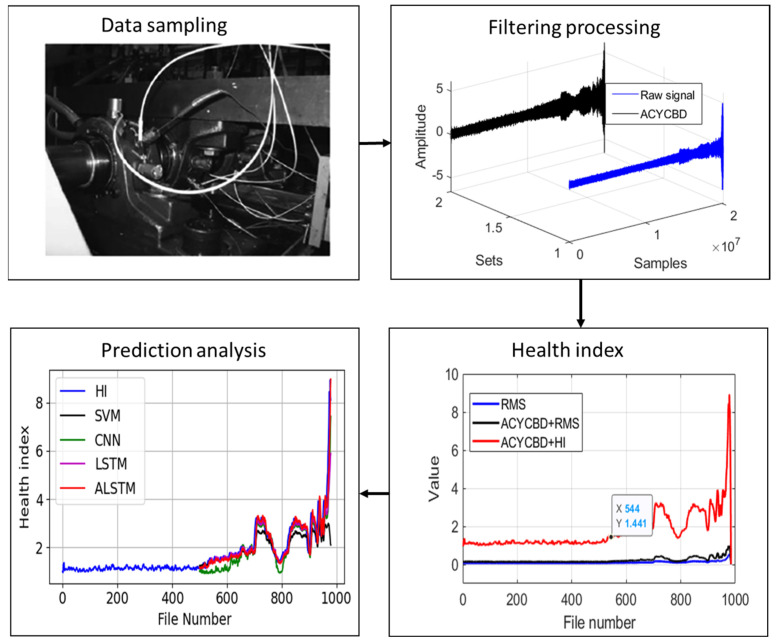
The proposed prognosis framework for whole approach processing.

**Figure 8 sensors-24-02382-f008:**
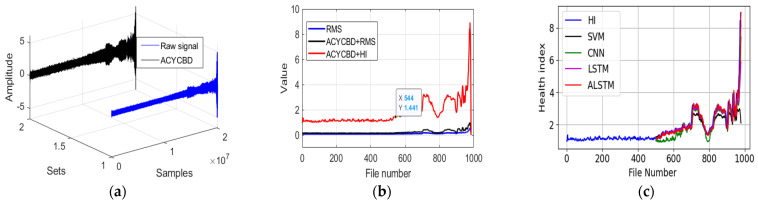
The proposed approach for the whole verified experiment: (**a**) raw and denoised signals, (**b**) RMS and HI analysis, and (**c**) the convolutional LSTM autoencoder prediction model.

**Figure 9 sensors-24-02382-f009:**
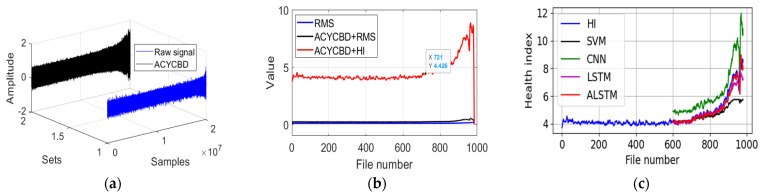
The proposed approach for the whole verified experiment: (**a**) raw and denoising signals, (**b**) RMS and health index analysis, and (**c**) the convolutional LSTM autoencoder prediction model.

**Figure 10 sensors-24-02382-f010:**
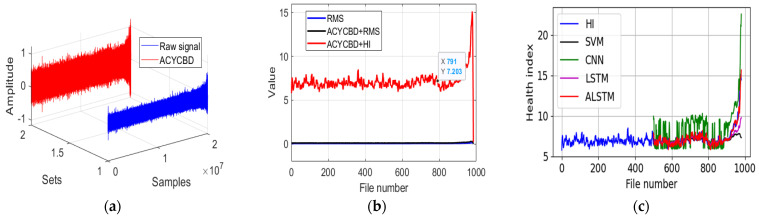
The proposed approach for the whole verified experiment: (**a**) raw and denoising signals, (**b**) RMS and health index analysis, and (**c**) the convolutional LSTM autoencoder prediction model.

**Table 1 sensors-24-02382-t001:** Parameters for the ALSTM architecture.

	Layer Type	Kernel/Stride
Encoding	Convolution	64 × 1/16 × 1
	Convolution	64 × 1/16 × 1
	LSTM	16 units
	Convolution	64 × 1/16 × 1
Decoding	Convolution	64 × 1/16 × 1
	LSTM	16 units
	Convolution	64 × 1/16 × 1
	Convolution	64 × 1/16 × 1

**Table 2 sensors-24-02382-t002:** Method prediction comparison.

RMSE	SVM	CNN	LSTM	GRU	DTGRU	ALSTM
Data1	0.88	0.56	0.54	0.44	0.44	0.42
Data2	0.93	1.18	0.43	0.56	0.49	0.35
Data3	1.15	1.80	0.88	0.83	0.63	0.52

## Data Availability

The data are available upon request.

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
