# Peer review of "The Prediction of the Remaining Useful Life of Rotating Machinery Based on an Adaptive Maximum Second-Order Cyclostationarity Blind Deconvolution and a Convolutional LSTM Autoencoder"

_sensors, 2024, doi:10.3390/s24082382_

Round 1
Reviewer 1 Report
Comments and Suggestions for Authors
The manuscript presents a novel approach for predicting the remaining useful life (RUL) of rotating machinery using adaptive maximum second-order cyclostationarity blind deconvolution (ACYCBD) and a convolutional LSTM autoencoder. The proposed method aims to improve feature extraction, health index analysis, and fault recognition for rotating machinery, ultimately enhancing RUL prediction accuracy.
The combination of ACYCBD and a convolutional LSTM autoencoder represents an innovative approach to RUL prediction in rotating machinery. By leveraging advanced signal processing and deep learning techniques, the proposed method offers a promising solution to the challenges associated with traditional RUL prediction methods. The authors conduct thorough experimental evaluations to validate the effectiveness of the proposed approach. Comparative analysis with SVM, CNN, and LSTM methods demonstrates superior performance in RUL prediction, indicating the potential practical utility of the proposed method. Please find some suggestions for improvement:
While the manuscript provides an overview of the ACYCBD and convolutional LSTM autoencoder algorithms, more detailed descriptions of the algorithms and their implementation specifics would enhance clarity, particularly for readers less familiar with these techniques. Besides, it is suggested to add more relevant work on the application of LSTM for remaining useful life prediction, such as: doi.org/10.1007/s40436-023-00464-y
The authors should discuss potential limitations and challenges associated with the proposed approach. Addressing factors such as computational complexity, data requirements, and real-world applicability would provide a more balanced assessment of the method's feasibility and practicality.
The interpretation of experimental results could be strengthened by providing insights into the implications of performance metrics, such as accuracy, precision, and recall, in the context of RUL prediction for rotating machinery. Additionally, visualizations or case studies illustrating the application of the proposed method in real-world scenarios would enhance the manuscript's impact.
The manuscript presents a novel and promising approach for RUL prediction in rotating machinery, leveraging advanced signal processing and deep learning techniques. With minor revisions addressing the identified suggestions for improvement, the manuscript has the potential to make a significant contribution to the field of machinery prognostics and health management.
Comments on the Quality of English LanguageThe language used throughout the manuscript is generally clear and concise.
Reviewer 2 Report
Comments and Suggestions for Authors
The proposed methodology integrates adaptive maximum second-order cyclostationarity blind deconvolution (ACYCBD) and a convolutional LSTM autoencoder for predicting remaining useful life in rotating machinery. While the overall paper is satisfactory, the authors should consider incorporating suggested enhancements to ensure its publication worthiness.
1. The abstract should emphasize the necessity and significance of the study in addressing challenges in predicting remaining useful life for rotating machinery.
2. In the introduction section, recent literature should be incorporated to provide context, and the limitations of previous studies should be discussed to highlight the novelty and effectiveness of the proposed method.
3. Ensure that equations used in Section 2 are properly cited.
4. Provide more detailed information about the experimental data used in the study and include a thorough analysis and discussion of the validation results to demonstrate the effectiveness and reliability of the proposed method.
5. In the conclusion section, include a discussion on future research directions and clearly outline the limitations of the current study to provide insights for further advancements in the field.
Comments on the Quality of English LanguageThe overall English in the manuscript is acceptable; however, minor revisions are needed.
Reviewer 3 Report
Comments and Suggestions for Authors
This paper introduces an RUL prediction method based on ACYCBD and a convolutional LSTM autoencoder. The research has a certain significance, however, some modifications are necessary to meet the requirements for acceptance. The detailed comments are as follows:
1. As far as I know, adaptive maximum second-order cyclistationarity blind deconvolution and convolutional LSTM autoencoders are both existing methods, so the authors should make it clear what original contributions are made during fusing these two methods.
2. In the introduction, the authors need to add some latest studies, such as transfer learning, incremental learning, small sample learning, etc. The following papers may be helpful.
Unsupervised Domain Deep Transfer Learning Approach for Rolling Bearing Remaining Useful Life Estimation, DOI: 10.1115/1.4062731
An Elastic Expandable Fault Diagnosis Method of Three-Phase Motors Using Continual Learning for Class-Added Sample Accumulations, DOI: 10.1109/TIE.2023.3301546
Meta-learning with Deep Flow Kernel Network for Few Shot Cross-domain Remaining Useful Life Prediction, DOI: 10.1016/j.ress.2024.109928
3. All the figures in the paper need to be improved. For example, Figure 1 can be removed, and the time domain and frequency domain diagrams in Figure 2 should be aligned.
4. In the experiments, the authors should compare their method with more new methods. In the manuscript, the existing comparison seems like ablation experiments not a regular comparison with existing methods.
5. Whether the samples used in the experiments are too few? In other words, how the generalization performance of the proposed method is.
Comments on the Quality of English Language
There are many grammatical errors in the paper, and the author should check them carefully.
Reviewer 4 Report
Comments and Suggestions for Authors
The article is focused and logical, using an ACYCBD filtering method to remove noise from the input signal and adopting the difference between the peaks and valleys as a health indicator, followed by the construction of an ALSTN hybrid model for the prediction of the remaining useful life of the bearings and its validation on three datasets.The innovativeness of the article is slightly lacking. Is there any improvement in the ACYCBD filtering method used, and where does it differ from previous work? In addition, the following issues should be further considered.
1. The acronym "PHM" appears several times in the text, so please standardize the use of this proper noun. The problem occurs in lines 30, 47, 58, 331, and 338.
2. During the interpretation of parameters in the text, parameter symbols need to be edited and further explained using formulas. For example, "s" in line 98, "N" in line 101 need to be edited, "𝑊, U, and b" in line 223, "RMS" in line 269, and "RMS" in line 223, "RMSE" in line 269, and "RMS" in line 280 need further explanation.
3. Please check the spelling of words in the text, such as "signal s" in line 136 and the related text in Figure 1.
4. Figure 4 needs further adjustment, including color matching and further modification of the text size and font.
5. The direction of the text in Figure 5 is suggested to be changed to bottom to top to make it easier for the reader to read.
6. Two datasets are used in line 283, but there is no clear description of the datasets.
7. Please compare the proposed method with the latest research techniques.
Comments on the Quality of English LanguageThe English shoud be further polished.
Reviewer 5 Report
Comments and Suggestions for Authors
Round 2
Reviewer 3 Report
Comments and Suggestions for Authors
The authors have revised their manuscript properly according to the comments. After revision, I think the manuscript can be accepted.
Author Response
We appreciate the positive response.
Reviewer 5 Report
Comments and Suggestions for Authors
Accept in present form
Author Response
We appreciate the positive response.